

# Accumulation and expression of horizontally acquired genes in *Arcobacter cryaerophilus* that thrives in sewage

Jess A. Millar and Rahul Raghavan

Biology Department, Portland State University, Portland, OR, United States

## ABSTRACT

We explored the bacterial diversity of untreated sewage influent samples of a wastewater treatment plant in Tucson, AZ and discovered that *Arcobacter cryaerophilus*, an emerging human pathogen of animal origin, was the most dominant bacterium. The other highly prevalent bacteria were members of the phyla Bacteroidetes and Firmicutes, which are major constituents of human gut microbiome, indicating that bacteria of human and animal origin intermingle in sewage. By assembling a near-complete genome of *A. cryaerophilus*, we show that the bacterium has accumulated a large number of putative antibiotic resistance genes (ARGs) probably enabling it to thrive in the wastewater. We also determined that a majority of candidate ARGs was being expressed in sewage, suggestive of trace levels of antibiotics or other stresses that could act as a selective force that amplifies multidrug resistant bacteria in municipal sewage. Because all bacteria are not eliminated even after several rounds of wastewater treatment, ARGs in sewage could affect public health due to their potential to contaminate environmental water.

## INTRODUCTION

Over the past few decades, based on numerous studies that examined the bacterial composition of wastewater during varying stages of treatment, there is growing evidence that sewage is an important hub for horizontal gene transfer (HGT) of antibiotic resistance genes (e.g., *Baquero, Martínez & Cantón, 2008*; *Zhang, Shao & Ye, 2012*; *Rizzo et al., 2013*; *Pehrsson et al., 2016*). Additionally, studies have shown that discharge of treated sewage allows these concentrated communities to spread into environmental water (*Okoh et al., 2007*; *Varela & Manaia, 2013*). The *Arcobacter* genus is commonly detected in sewage treatment plants around the world (*Collado et al., 2008*; *Zhang, Shao & Ye, 2012*; *Varela & Manaia, 2013*). This sparsely studied Epsilonproteobacteria is frequently associated with veterinary diseases, and is closely related to *Campylobacter*, and is considered an emerging human pathogen that causes enteritis and bacteremia (*Kabeya et al., 2004*; *Morita et al., 2004*; *Collado et al., 2008*). In addition, *Arcobacter* is known to be resistant to a wide array of commonly used antibiotics, with varying resistance profiles observed in different species (*Houf et al., 2004*; *Abay et al., 2012*; *Rahimi, 2014*), but the genes that enable antibiotic resistance are mostly unknown (*Abdelbaqi et al., 2007*; *Miller et al., 2007*).

Corresponding author
Rahul Raghavan,
rahul.raghavan@pdx.edu

In this study, we examined the bacterial diversity and the presence and expression of antibiotic resistance genes (ARGs) and virulence factors in untreated sewage. Our analyses revealed that an *A. cryaerophilus* strain that contained multiple putative ARGs was a major constituent of the sewage microbiome. In addition, we observed that a large number of ARGs and virulence factors were expressed in *A. cryaerophilus* and in the rest of the sewage microbiome, which portends potential public health risk if bacteria carrying these genes contaminate public water resources.

## MATERIALS & METHODS

### Sewage sample collection, DNA and RNA extraction, and deep-sequencing

Three untreated sewage influent samples (50 ml) were collected at the Roger Road Wastewater Reclamation Facility, Tucson, Arizona (March 2012) and immediately transferred on ice to the laboratory and stored at −80 °C until further use. The sewage samples were spun down (12,000 × G, 15 min, 4 °C) and the pellets were suspended in 1 ml of TRI reagent (Life Technologies). Total RNA from each sample was extracted from the aqueous phase and corresponding DNA was isolated from the interphase using a protocol provided by the manufacturer. RNA samples were treated with TurboDNAse (Life Technologies) to remove contaminating DNA, and PCR reactions using 16S rDNA primers were performed to confirm complete DNA removal. Furthermore, RNA samples were depleted of ribosomal RNA using RiboZero Bacteria and RiboZero Human kits (Illumina). Around 100 $\eta$g of RNA from each sample was used to prepare directional mRNA-seq libraries using the Illumina Small RNA Sample Preparation Kit and Directional mRNA-seq Sample Preparation protocol provided by Illumina Inc. DNA samples were further purified using DNeasy kit (Qiagen) and around 5 µg of DNA from each sample was used to prepare paired-end DNA-seq libraries using the Paired-End sample Preparation Kit (Illumina). All RNA libraries were pooled into a single lane, and all DNA samples were pooled into another lane of Illumina HiSeq 2000 and were sequenced at the Yale Center for Genome Analysis (RNA-seq: single-end, 75 cycles; DNA-seq: paired end, 2 × 75 cycles) using standard adapters. All DNA and RNA reads have been deposited at NCBI (BioProject PRJNA354077).

### Taxonomic classification

DNA reads were cleaned by removing adapters and were filtered by quality ($\geq$Q20) and length ($\geq$50 bp) using Trimmomatic v0.32 (*Bolger, Lohse & Usadel, 2014*). Each library was assembled into contigs using 17,000,000 reads and IDBA-UD (*Peng et al., 2012*), and normalized to the smallest library size (6,000 contigs) using Seqtk (https://github.com/lh3/seqtk) (*Perner et al., 2014*) (Table S1). All contigs were searched against the NCBI nt database using BLAST and analyzed in MEGAN (*Huson et al., 2007*), requiring at least 70% of the query sequence to align with the subject sequence with $\geq$70% identity to be assigned to a given phylum. Contigs classified at the phylum level (48%, 57%, and 54% of contigs from the three samples, respectively) were used to determine their detailed taxonomic positions. Remaining contigs either did not have significant BLAST

hits or mapped to unidentified environmental samples; however, all contigs were used in downstream analyses (detection of antibiotic resistance genes, virulence factors etc.).

## Gene expression and identification of antibiotic resistance genes, virulence factors, transposases, and bacteriophage genes

Contigs were run through MetaProdigal (*Hyatt et al., 2012*) to identify encoded ORFs, which were annotated by mapping to antibiotic resistance genes, virulence factors, bacterial transposases, and prophages obtained respectively from CARD, PATRIC, InterPro, and PHAST databases using PHMMER, with an $E$-value of at least 1e−10 and percent identity greater than 25% (*Finn, Clements & Eddy, 2011*; *Zhou et al., 2011*; *McArthur et al., 2013*; *Wattam et al., 2014*; *Mitchell et al., 2015*). For determining presumptive ARGs, we further performed BlastX using the ORFs as queries and the respective CARD proteins as subjects to determine the percentage coverage values. Table S7 provides the $E$-value, percent identity and percent coverage for each potential ARG detected in our *A. cryaerophilus* contigs.

RNA reads from each sample were filtered by quality ($\geq$Q20) and length ($\geq$50 bp) using Trimmomatic v0.32 and normalized using Seqtk to 50,000,000 reads. They were mapped to annotated ORFs using CLC Genomic Workbench v6.5. A strict mapping criterion (at least 95% of each read should map with at least 95% identity to the mapped region) was used in order to minimize non-specific mapping. Genes were filtered and considered expressed based on at least 10 reads mapping to each ORF. Statistical analysis was conducted using SAS Studio v3.4 (SAS Institute, Cary NC).

### *Arcobacter* genome assembly

DNA reads from all three samples were pooled to gain enough coverage depth, and were assembled into contigs using IDBA-UD (*Peng et al., 2012*). All contigs were searched against the NCBI nt database using BLASTN and analyzed in MEGAN (*Huson et al., 2007*). All *Arcobacter* gene sequences were downloaded from NCBI, and using PHMMER, *Arcobacter* contigs present in our data were identified with at least 1e−10 $E$-value as the cutoff. These contigs were extracted and run through the differential coverage binning procedure for metagenomic data, as described previously (*Albertsen et al., 2013*). In brief, contigs were binned based on coverage, tetranucleotide frequency, GC%, and length, and then examined for presence of essential single copy genes. Phylogenetic analyses were conducted on nucleotide sequences using several housekeeping genes to identify the bins containing *A. cryaerophilus* (marked in blue in Fig. S1). One genome bin with ∼200× coverage that contained all *A. cryaerophilus* housekeeping genes was selected for secondary refinement and finishing (top right cluster in Fig. S2). This cluster of contigs was isolated and all original trimmed DNA reads were mapped against them using Bowtie2 v2.1.0 (*Langmead & Salzberg, 2012*). All mapped reads were then reassembled into contigs using IDBA-UD. These contigs were combined with all original trimmed DNA reads for scaffold extension using SSPACE (*Boetzer et al., 2011*) into a final scaffold of ∼1.8 Mb over 456 contigs. To check for completeness of the assembled *A. cryaerophilus* genome, we used a single-copy gene database (*Albertsen et al., 2013*), and as a control we performed the same analysis with the *A. butzleri* (CP000361.1) genome. Visual representation of draft genome was created

using Circos (*Krzywinski et al., 2009*). The draft genome has been deposited at NCBI under the accession LNTC00000000.

## Detection of HGT

Horizontally acquired genes were identified using HGTector (*Zhu, Kosoy & Dittmar, 2014*). *Arcobacter* was set as self-group, and Campylobacterales was set as exclusion group. This method captured HGT events where only *Arcobacter* has acquired a particular gene from outside of Campylobacterales and ignored any events where the genes could also have been transferred elsewhere within the order. This conservative approach was used due to the dearth of annotated genomes within Campylobacterales. BLASTN parameter thresholds were set at 70% identity and an *E*-value of at least 1e−5. Several putative HGT genes were examined using phylogenetic analysis to validate the HGTector data (Fig. S3).

## Phylogenetic analysis

Nucleotide sequence alignment for all trees was performed using Clustal Omega (*Sievers et al., 2011*), and ambiguously aligned regions were removed using Gblocks (*Talavera & Castresana, 2007*). The evolution model GTR+I+G (General Time Reversible plus Invariant sites plus Gamma distribution) used for all trees was selected using jModelTest2 (*Darriba et al., 2012*). Bayesian trees were constructed using MrBayes as implemented in Geneious (*Huelsenbeck & Ronquist, 2001*; *Kearse et al., 2012*). A chain length of 1,000,000 was used with a burn-in fraction of 25% and sampling every 100 trees. Maximum Likelihood trees were constructed using RAxML (*Stamatakis, Hoover & Rougemont, 2008*) as implemented in Geneious with 1,000 bootstrap replicates to confirm Bayesian topologies. *Helicobacter pylori* (AJ558222.1) was used to root all phylogenetic trees.

# RESULTS AND DISCUSSION

## *A. cryaerophilus* thrives in sewage

For the three sewage samples, taxonomic labels were assigned to at least the phylum level for all contigs with significant BLAST hits. There was no significant difference in bacterial distribution between the three samples (Fig. 1); hence, average values are presented hereafter. Members of the phylum Proteobacteria (67% of total hits) was the most prevalent bacteria, followed by Bacteroidetes (23%) and Firmicutes (9%). A more comprehensive study that examined several sewage samples from across the USA observed a similar pattern of bacterial phyla abundance (*Shanks et al., 2013*). However, at the genus level, *Arcobacter* (an Epsilonproteobacteria) was the most dominant bacterium in our study, making up 39% of all annotated contigs, unlike members of Gammaproteobacteria (38% of all pyrotags) in the previous study (*Shanks et al., 2013*).

*Arcobacter* is commonly associated with both humans and farm animals (*Collado et al., 2008*), the latter perhaps more relevant to this specific wastewater treatment plant because agriculture accounts for the largest use of water at around 70% of all water demand within the state of Arizona (*ADWR, 2009*). In addition, in Tucson, the wastewater treatment plant served both agricultural and municipal areas (*PAG, 2006*). Members of the phyla Bacteroidetes and Firmicutes, two of the most abundant bacteria in human gut, were
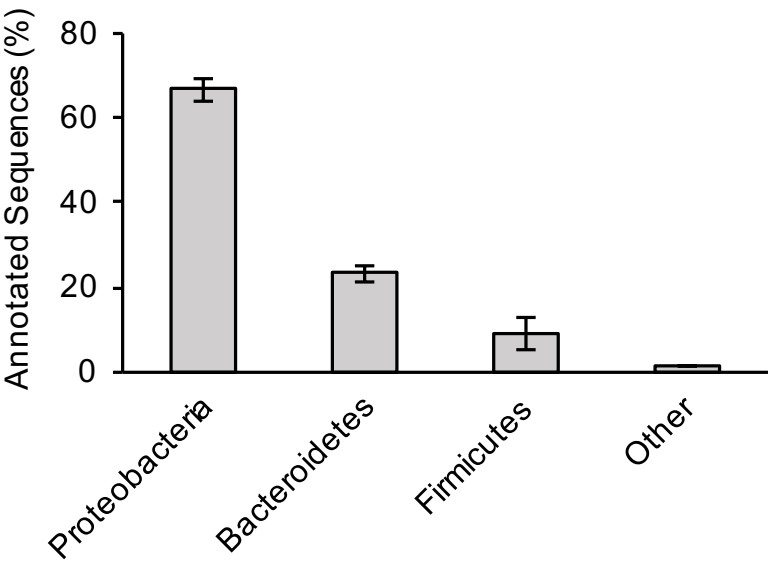

**Figure 1** **Bacterial composition of sewage samples.** Percentage of sewage contigs that were assigned to each bacterial phylum is shown. Data represent means of three samples ± standard deviations.

also abundant in the sewage samples (*Cho & Blaser, 2012*; *Jandhyala et al., 2015*). Taken together, these data highlight the important role that sewage systems play as an arena where bacteria of human and animal origin interact, which could promote the exchange of genes between the two groups (*Baquero, Martínez & Cantón, 2008*; *Gaze et al., 2013*; *Rizzo et al., 2013*). After conducting genomic binning of the *Arcobacter* contigs utilizing various factors such as coverage depth, GC%, and tetranucleotide frequency (*Albertsen et al., 2013*), we were able to identify ∼80% of these contigs as belonging to *A. cryaerophilus*, an emerging human pathogen that is commonly associated with diseases such as bovine reproductive disorders, diarrhea and hemorrhagic colitis in cattle and sheep (*Schroeder-Tucker et al., 1996*; *Ho, Lipman & Gaastra, 2006*).

**Presence and expression of putative ARGs in *A. cryaerophilus***

To better characterize *A. cryaerophilus*, we assembled a near-complete genome from the DNA-seq reads (Fig. 2). Based on the presence of 100 out of 106 single copy genes (*Albertsen et al., 2013*) with zero redundant copies, we estimate that the *A. cryaerophilus* genome is ∼95% complete and contains 2,419 ORFs (including partial genes at the ends of contigs) (Table S2). Among these ORFs, 115 (5% of ORFs) encode putative antibiotic resistance genes (ARGs) belonging to 25 categories as defined by the CARD database (Tables S3 and S7) (*McArthur et al., 2013*). Macrolide resistance made up the majority of annotated ARGs (26, 23%,) (Table S3), with fluoroquinolones (18, 16%), aminocoumarin (17, 15%) and vancomycin (13, 11%) resistance genes being the next largest groups. Because gene expression is a good representation for functional gene activity, we analyzed the expression of *A. cryaerophilus* genes using RNA-seq and discovered that all 115 presumptive ARGs genes were expressed (Fig. 3A; Data Set S1). In comparison, *Helicobacter pylori*, a closely related Epsilonproteobacteria contain 59 ARGs (4% of genes); however, in both bacteria

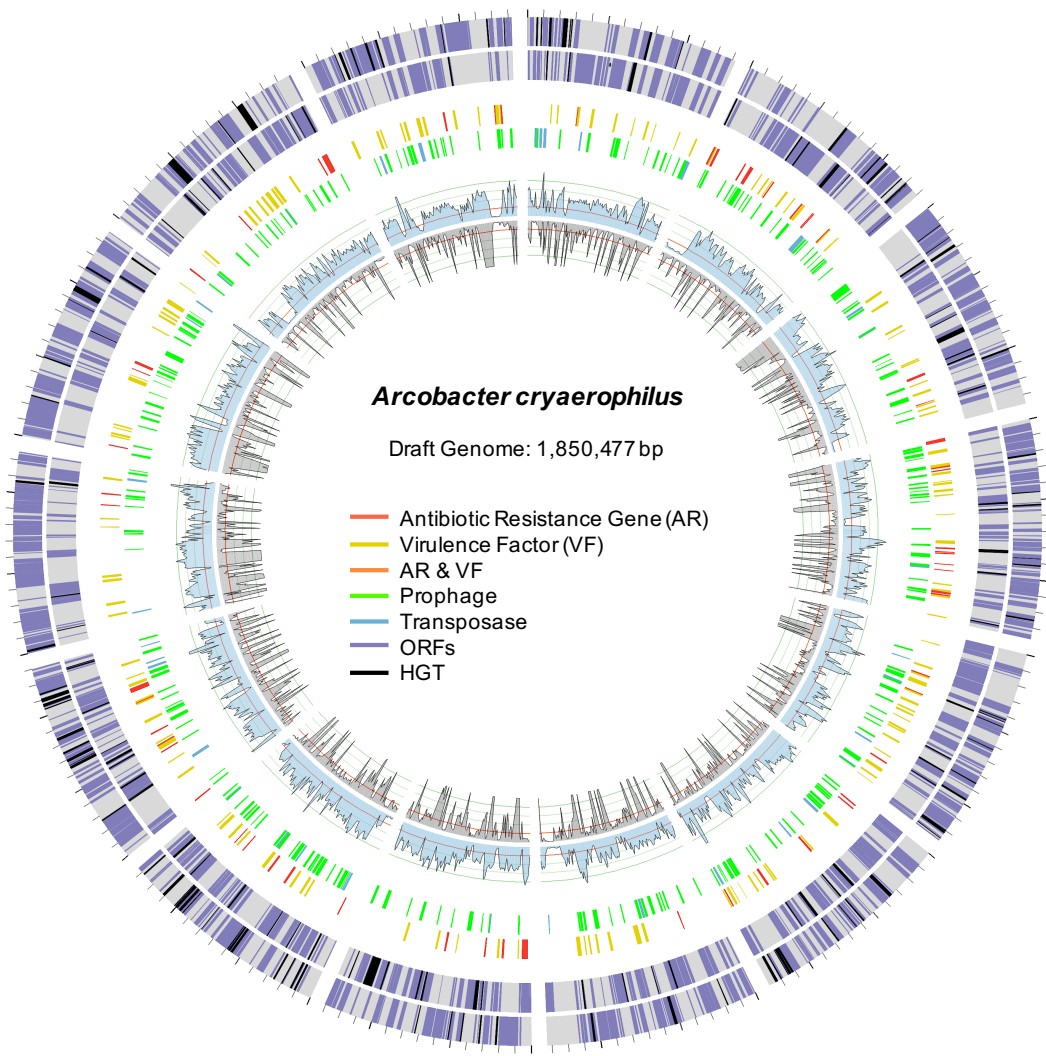

**Figure 2** **Draft genome of *A. cryaerophilus*.** Two outer rings show ORFs (purple) on forward and reverse strands, respectively. Black blocks represent horizontally acquired genes. Each tick mark represents 10,000 bp. Middle two rings show positions of features annotated in the center. Inner blue and grey rings show DNA-seq coverage (mean of three samples) and RNA-seq transcription levels (mean of three samples), respectively. Note that 456 original contigs were randomly assigned to 14 equal fragments for easy visualization.

around 50% of ARGs consisted of efflux pumps (*Paulsen, Sliwinski & Saier, 1998*). It should be noted that some of the candidate ARGs identified in this study are possibly genes with other functions but were detected because they have homology to known ARGs; additionally, although all ARGs were found to be expressed in *A. cryaerophilus*, the median level of expression of single copy genes (3,887 reads mapped) was found to be 10× higher than the median level of expression of ARGs (363 reads mapped), probably because higher expression of many ARGs requires strong induction.
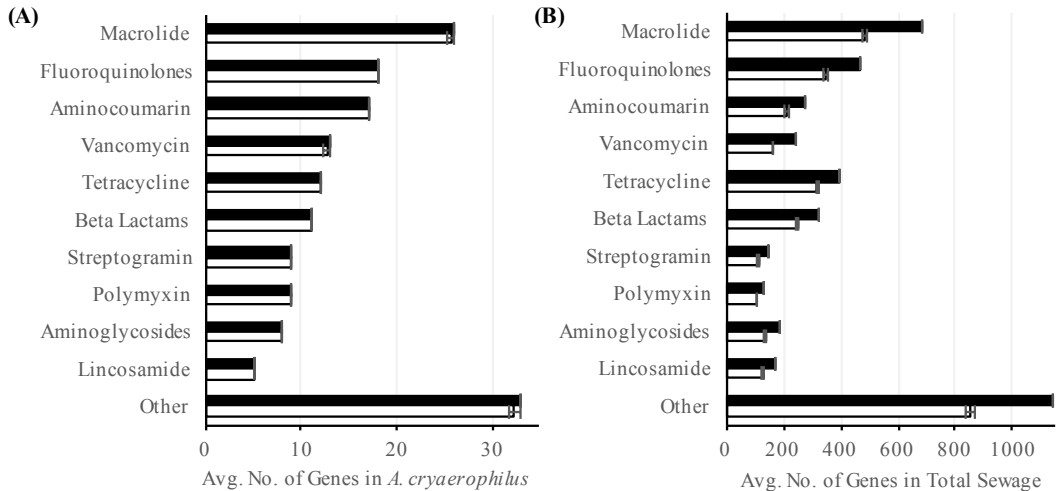

**Figure 3** **Abundance of putative antibiotic resistance genes.** Number of annotated (black) and expressed (white) antibiotic resistance categories in (A) *A. cryaerophilus* and (B) total sewage sample is shown. Data represent means of three samples ± standard deviations.

To determine the prevalence of potential ARGs in the total sewage, we extended our analysis to all contigs assembled in our study. Out of the 60,723 ORFs encoded in the sewage contigs, 2,606 ORFs matched 42 antibiotic resistance categories (Table S4). Using RNA-seq, we determined that 2,106 (81%) of these ORFs were expressed (Fig. 3B). Of the 2,106 putative antibiotic resistance genes expressed in the sewage samples, macrolide resistance genes made up the largest portion (538, 26%) (Table S4). The next two largest groups were fluoroquinolone resistance (378, 18%) and tetracycline resistance (339, 16%) genes. The expression of likely ARGs could be due to the presence of numerous antibiotics in urban wastewater (*Heberer, 2002*; *Rizzo et al., 2013*), which could select for multidrug resistant bacteria, thereby aggravating an already dire situation (*Baquero, Martínez & Cantón, 2008*; *Rizzo et al., 2013*; *Wellington et al., 2013*; *Amos et al., 2014*)). It is also possible that ARGs were being expressed constitutively or in response to stress (*Poole, 2012*). Additionally, previous studies have shown that ARGs are expressed in a wide variety of environments even in the absence of known anthropogenic antibiotic pressure (*Udikovic-Kolic et al., 2014*; *Versluis et al., 2015*; *Noyes et al., 2016*); hence, further study is required to determine the stimuli for the observed ARG expression. Interestingly, although the total sewage contigs contained 23× more ARGs than in *A. cryaerophilus* contigs (2,606 vs. 115), 44% of DNA reads mapped to *A. cryaerophilus* ARGs, indicating that while the sewage contained high diversity of ARGs, most non-*A. cryaerophilus* ARGs were of low abundance.

## Signatures of HGT in *A. cryaerophilus* genome

We compared our draft genome of *A. cryaerophilus* to the published genome of *A. butzleri* (CP000361.1), a closely related human and animal pathogen that has been studied much more extensively than *A. cryaerophilus* (*Vandenberg et al., 2004*; *Miller et al., 2007*; *Collado et al., 2008*) (Fig. S4). As observed previously in other members of this genus (*Karadas et*
**Table 1** Comparison of *Arcobacter* genomes.

| Features | *A. cryaerophilus* | *A. butzleri* | Overlap |
|---|---|---|---|
| Total ORFs | 2,419[a] | 2,259 | 1,337 |
| Horizontally Acquired ORFs | 209 | 228 | 73 |
| Antibiotic Resistance | | | |
|     Categories | 25 | 29 | 23 |
|     Genes | 115 | 140 | 54 |
| Virulence Factors | | | |
|     Categories | 24 | 24 | 22 |
|     Genes | 232 | 185 | 92 |
| Transposases | | | |
|     Categories | 7 | 7 | 5 |
|     Genes | 61 | 57 | 15 |
| Prophages | | | |
|     Genes | 290 | 320 | 173 |
| GenBank Accession | LNTC00000000 | CP000361.1 | |

**Notes.**
[a] Includes partial genes at the ends of contigs.

*al., 2013*; *Merga et al., 2013*), the two *Arcobacter* species only shared 1,337 genes (∼50%) (Table 1, Data Set S2). A comparison of the two genomes was also conducted using RAST (*Overbeek et al., 2014*), which showed that merely 846 genes with known functions were shared between *A. butzleri* and *A. cryaerophilus* (Data Set S3). This sizable variation in gene content between the two species indicates that HGT could have played a prominent role in shaping the genomes of *Arcobacter* species. Concomitantly, even after using a very conservative threshold, we detected 209 (9%) and 228 (10%) horizontally acquired genes in *A. cryaerophilus* and *A. butzleri*, respectively (Table 1). While similar in scale, only 73 HGT-origin genes were shared between the two genomes, indicating that parallel HGT events have molded the genomes of the two *Arcobacter* species.

HGT is known to promote ARG dissemination between bacteria (*Hawkey & Jones, 2009*; *Gaze et al., 2013*; *Pehrsson et al., 2016*); hence, we compared the presumptive ARGs present in *A. cryaerophilus* to those present in *A. butzleri* in order to identify those that are of possible HGT origin. We identified 140 putative genes belonging to 29 antibiotic resistance categories in *A. butzleri*, and out of the 25 antibiotic resistance categories present in *A. cryaerophilus,* 23 were present in *A. butzleri*, with two categories (Glycylcycline and Roxithromycin resistance) found only in *A. cryaerophilus*, and six categories (Bicyclomycin, Elfamycin, Isoniazid, Kanamycin, Streptomycin, and Teicoplanin resistance) exclusive to *A. butzleri* (Table 1). However, within each category large differences in gene content was observed between the two bacteria, with only 54 genes shared between *A. cryaerophilus* and *A. butzleri*. These data show that even though the antibiotic resistance capabilities of both bacteria overlap, their respective gene repertoires were largely assembled through independent HGT events. Transposons and bacteriophages are important agents of HGT in bacteria, and we found several transposases and bacteriophage ORFs in *A. cryaerophilus* (61 transposase ORFs, 290 phage ORFs) and *A. butzleri* (57, 320) (Fig. 2 and Table 1, Table S2).

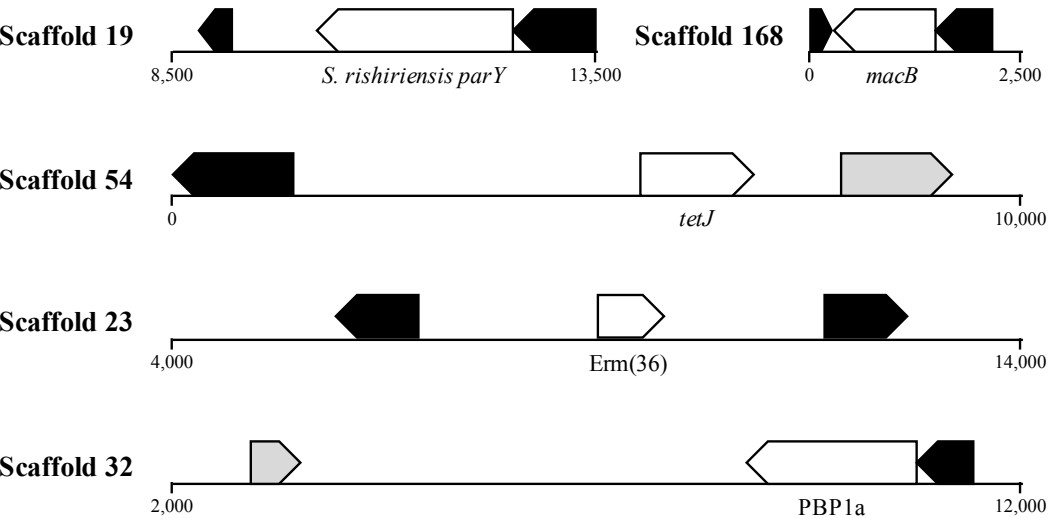

**Figure 4 Location of ARGs indicates horizontal acquisition.** Several potential antibiotic resistance genes (white) in *A. cryaerophilus* that are flanked by prophage genes (black) and transposases (grey) are shown. Nucleotide positions within each contig are also provided.

Additionally, we discovered that several of the candidate ARGs in *A. cryaerophilus* were located in close proximity to transposases or bacteriophage genes (Fig. 4), suggestive of a role for these mobile genetic elements in the accumulation of ARGs in this emerging pathogen.

## Presence and expression of virulence factors in sewage

In addition to ARGs, another class of genes in *A. cryaerophilus* that could potentially impact human health is virulence factors. Most of the previous work at the molecular level has focused on nine putative virulence genes first described in *Arcobacter butzleri* strain RM4018. The presence of these nine virulence genes in *Arcobacter* genomes is highly variable and are all rarely found together in the same genome (*Miller et al., 2007*; *Douidah et al., 2012*). In general, the ability to adhere to and invade cells varies widely between *Arcobacter* species, with some of the most invasive strains isolated from feces or sewage samples (*Ho et al., 2007*; *Karadas et al., 2013*; *Levican et al., 2013*). Using the PATRIC database we identified 232 putative virulence genes (24 virulence categories) in *A. cryaerophilus* (Table S5), out of which 231 were expressed (Data Set S4). In PATRIC, virulence factors are assigned the category "virulence" if their mode of action is not specified in an associated study. Among the expressed virulence factor genes, 101 were annotated with a category other than "virulence." Of these, "intracellular survival and replication" was the largest group (30, 30%) (Table S5). The next largest groups present were "cellular metabolism" (22, 22%), "adhesion" (18, 18%), and "invasion" (11, 11%) (Fig. 5A). In the total sewage contigs, we identified 4,440 putative virulence factor genes (38 virulence categories (Table S6), out of which, 3,776 were expressed (Data Set S4)). Excluding the "virulence" category, 1,812 genes belonging to 37 other virulence categories were identified in the sewage microbiome. Of these, 1,589 genes from 35 categories were

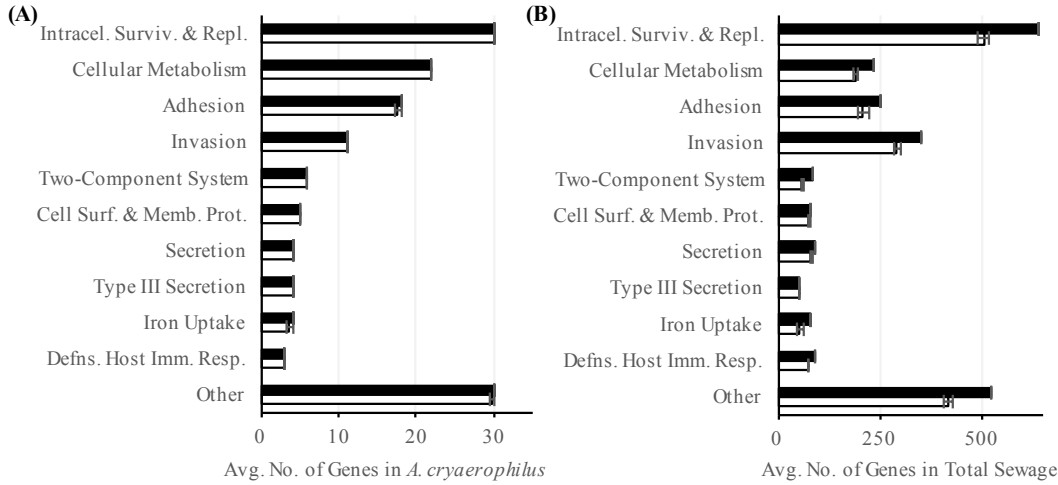

**Figure 5 Abundance of virulence factors.** Number of annotated (black) and expressed (white) virulence factor categories in (A) *A. cryaerophilus* and (B) total sewage sample is shown. Data represent means of three samples ± standard deviations.

expressed, with "intracellular survival and replication" (548, 35%), "invasion" (318, 20%) and "adhesion" (229, 14%) being the top three categories (Fig. 5B; Table S6).

Our data suggest that untreated sewage contains several genes that potentially promote bacterial antibiotic resistance and virulence, and that *A. cryaerophilus*, a potential human pathogen that contains multiple drug resistance and virulence factors is a major component of this sewage system. Because we analyzed only a limited number of samples, further study is required to determine whether the dominance of *A. cryaerophilus* was a short-term phenomenon or whether this bacterium is a long-term resident of this sewage system (*McLellan et al., 2010*; *Shanks et al., 2013*). Although its cause is not understood, as observed in our study, *Arcobacter* has been shown to be highly prevalent in other sewage systems (*Fisher et al., 2014*). A possible explanation is the formation of biofilm on pipe surfaces and in deposited sediments along the sewer system (*Chen, Leung & Hung, 2003*), another possibility is that the presence of multiple antibiotics, heavy metals or xenobiotics in wastewater, even at very low concentrations is selecting for *A. cryaerophilus*, which contains multiple ARGs (*Heberer, 2002*; *Hawkey & Jones, 2009*; *Gullberg et al., 2014*; *Jutkina et al., 2016*). Similar to our observation, selection for antibiotic resistant bacteria has been described from other wastewater treatment plants (*Goñi Urriza et al., 2000*; *Czekalski et al., 2012*; *Mao et al., 2015*); consequently, constant monitoring of both pre- and post-treatment sewage is warranted because of the risk of reintroducing bacteria replete with ARGs and virulence factors into natural environments (*Fahrenfeld et al., 2013*; *Czekalski, Gasco & Burgmann, 2014*; *Mao et al., 2015*; *Pehrsson et al., 2016*).

## ACKNOWLEDGEMENTS

We thank Howard Ochman for collecting the sewage samples and Peter King for assistance with data analysis.

### Funding

This work was supported by Portland State University. The funders had no role in study design, data collection and analysis, decision to publish, or preparation of the manuscript.

### Grant Disclosures

The following grant information was disclosed by the authors:
Portland State University.

### Competing Interests

The authors declare there are no competing interests.

### Author Contributions

- Jess A. Millar conceived and designed the experiments, performed the experiments, analyzed the data, wrote the paper, prepared figures and/or tables, reviewed drafts of the paper.
- Rahul Raghavan conceived and designed the experiments, analyzed the data, wrote the paper, reviewed drafts of the paper.

### DNA Deposition

The following information was supplied regarding the deposition of DNA sequences:
GenBank: LNTC00000000 and PRJNA354077.

### Data Availability

The raw data has been supplied as a Supplementary File.

### Supplemental Information

Supplemental information for this article can be found online at http://dx.doi.org/10.7717/peerj.3269#supplemental-information.

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
