# Peer review of "Accumulation and expression of multiple antibiotic resistance genes in Arcobacter cryaerophilus that thrives in sewage"

_PeerJ, doi:10.7717/peerj.3269_

## Round 0.1 · original submission · Major Revisions

Dear Rahul and Jess,

I thank both of you for submitting your work to PeerJ, and the two reviewers for putting their valuable time and effort into evaluating it.

Although the study clearly has merit for publication, both reviewers have concerns regarding the methods used, and I agree with their assessment of these shortcomings.

It is important to provide much more information about the parameters used for quality filtering, assembly, and mapping, as well as detailed reports regarding the results of these steps. Among others, these parameters should include details such as the minimum alignment length and identity required for mapping. As a result of the use of very short reads in this study, makes the stringency of mapping much more critical to minimize non-specific mapping of transcripts.

The binning is an extremely important in the context of your study, however, it is described very poorly. As one of the reviewers point out, it is not clear why an initial phylogenetic binning is used (i.e., recruiting contigs that contain Arcobacter genes), when the coverage-based binning is superior to phylogenetic binning, especially in environments that lack comprehensive representation in databases.

Although the number of single-copy core genes found in the genome bin is mentioned, the redundancy is not reported. Please consider using CheckM or any other tool that provides relevant estimates for genome bins to make sure completion/contamination values are reported from multiple sources. It would also be useful to include a visualization of this genome bin, where the coverage of each contig is seen. A study that discusses HGTs should be even more stringent and clear about how it addresses contamination in their bins.

The methods section contains rather vague statements such as this one: "One genome bin with ~200x coverage that contained all A. cryaerophilus housekeeping genes was selected for secondary refinement and finishing". What is exactly secondary refinement and finishing? How was the refinement done? The purpose of the methods section is to make sure the study is reproducible, however, the methods section in this study does not give me any impression that I can reproduce what I read.

If you were to submit a revised manuscript, please address each comment made by each reviewer in your rebuttal.

Thank you very much,

·

Basic reporting

my comments are combined in the general comments section below

Experimental design

my comments are combined in the general comments section below

Validity of the findings

my comments are combined in the general comments section below

Additional comments

in their manuscript "Accumulation and expression of multiple antibiotic resistance genes in Arcobacter cryaerophilus that thrives in sewage" the authors describe sequencing of wastewater influent, binning of a draft genome of arcobacter cryoaerophilus and analysis of antibiotic resistance genes in the binned genome and total dataset.

The subject of the study is interesting, but the description of the methods lacks details and needs to be substantially expanded before the manuscript can be published. The molecular biology section is merely a list of kits, the library preparation protocol is not mentioned, sequencing output is not listed and in almost all cases paramaters of used software are absent.

I am not convinced by the authors "10 read" criterion for expression of ARGs, as more sequencing will eventually pick up reads from a given sequence. Additionally, the authors' choice to only map to the annotated ORFs might lead to substantial false positives by "forcing" reads onto related ORFs. This is not a problem if very stringent mapping parameters were used, but these parameters are not provided...

It is unclear to me why the authors do a phylogenetic binning prior to their differential coverage approach. As such, any contigs that are part of the arcobacter genome, but contain a novel fraction of the arcobacter pan-genome are automatically excluded from the draft genome. Since completeness estimates only factor in the core genome, I am skeptical of the completeness of the presented genome. That said, 1.8 Mb is relatively close to the published 2.1 Mb Arcobacter cryoaerophilus genome, so I might be overly skeptical here. Could the authors include for example an alignment of closely related genomes?

specific points:

- line 23: "which are major constituents of human gut microbiome ..." Even though this is partially human waste, Firmicutes and Bacteroidetes are large and diverse groups. Did you actually detect constituent of the microbiome? If not rephrase to eg "bacteroides and firmicutes, member of which are ..."

- line 57: Were DNA/RNA and the libraries quality checked at any point?

- line 71: was normalization done after assembly? If so, why?

- line 71: please provide some assembly statistics, such as contig numbers, length etc. to provide some context for interpretation of the complexity of the communty.

- line 72: BLASTn isn't a very good tool for taxomonic assignment of long nucleotide sequences, as all hits to a single gene will likely be shown first. How well was the length of the nt BLAST hit correlated with the length of the contig? IN other words, were longer contigs assigned to a certain taxonomy based on only a fraction of their length?

- line 157: the number of genes for a 1.8 Mb genome seems really high. I usually expect a number close to 1 gene per 1000 bases, so 1800-2000 genes for a genome this size. As the genome is highly fragmented, have you checked whether many of the genes are fragmented? (and possibly represent halves of the same gene?)
The 6 other arcobacter cryoaerophilus genome assemblies (see: https://www.ncbi.nlm.nih.gov/genome/genomes/11530) have between 2082 and 2352 predicted genes on genomes of 2.0-2.3 Mb

·

Basic reporting

English is generally good and the MS well written, but several small errors throughout that need correcting.

References are generally good, but I have noted several instances where relevant references should be added.

Figures (in text and in supplement) need more detail in the captions to be fully appreciated.

In several places throughout the text, discussion of the animal origin of Arcobacter in sewage is mentioned, but no real hypothesis for how animal-origin bacteria got into sewage is put forth. This seems irrelevant to the rest of the narrative. I would recommend clarifying the focus of Arcobacter and its origin/presence/relationship to humans and urban infrastructure.

Two general comments from the introduction:
(line 40 and throughout) Be careful with the use of “Arcobacter” or “this bacterium” it is a genus that contains many species. Be clear when you are referring to the genus vs. the species A. cryaerophilus vs. the assembled genome

Different Arcobacter spp. have varying resistance profiles to different antibiotics (lines 45-47)

Experimental design

The methods need to be described in better detail to accurately capture the work that was done, both for the purpose of understanding the validity of the findings and to be able to replicate.

Specific comments regarding the methods:
1 mL is a very small volume for extraction of DNA/RNA since sewage is highly diluted (vs a stool sample). A more typical method is filtration of 10-50 mL and preservation of filters (e.g., CITE) or centrifugation and extraction from a pellet. What were the DNA and RNA yields from these extractions?


Please provide more details on the sample preparation and sequencing method. E.g., how was removal of DNA confirmed for RNA samples? What type of sequencing kit/chemistry was used for RNA-seq? for DNA? Were all samples sequenced on a single run? Standard adapter

Please provide a breakdown of sequence data (supplementary table would be appropriate): how many reads were generated for each DNA and RNA sample? How many of these passed quality? How many were associated with the genus Arcobacter ?

Please check your NCBI submission info: PRJNA322050 did not bring up any information when I searched (line 66); the other numbers provided not in the text but on the review site did link to sequences, so please update in the text if those are correct.

Why were the three libraries normalized? They are essentially technical replicates from the same sample (if not, clarify that in the sample collection methods), so wouldn’t you want all of the data possible?

Please provide information (again, a supplementary table would be fine) with the general results of the contig assembly (what was the range of contig sizes? Average?) for each sample (DNA and RNA).

Please clarify the taxonomic classification if I am understanding this incorrectly: phylum level bins were set up for all contigs based on NCBI hits? I don’t understand how contigs were annotated at the phylum level? Did they have equal similarity matches from several different organisms? Or did you just take everything from a given phylum that had a significant match and put them all in that bin? What was the % similarity criteria used to be considered a match? Were the remaining contigs used in the downstream analyses?

(Line 84) again, it would be interesting to know how many quality sequences were initially generated before normalization

Please provide information on quality filtering and contig assembly for RNA. Same as DNA? This is not mentioned anywhere.

(Lines 89-106) Please add a supplementary table summarizing the Arcobacter contigs (length of contigs, coverage).

(lines 99-101) If all DNA reads were mapped back to the A. cryaero. contigs, it is possible that some of the reads from the other Arco genomes mapped back to A. cryaerophilus? All sewage systems are somewhat unique in their Arcobacter composition, but most have several relatively abundant species. Many cities also have two different strains of A. cryaerophilus, one of which is of human clinical significance (see Fisher 2014 in your refs). A tool like Anvi’o might be better at sorting out if these sequences are really from one strain or from multiple strains and possibly even different species. Was any follow up work done with the other Arcobacter sewage contig that did not group with A. cryaerophilus?

(line 106) LNTC000000000 does not show up as anything in NCBI; please check the submission info for the draft genome.

(Lines 117-126) Somewhere in the methods or in the supp. Figs. (caption?) include the nt length for each housekeeping gene. Also, those trees need some cleanup, even though they are in Supp. Info – italicize species names, make the branch bootstrap numbers not overlap the tree structure, etc.

Validity of the findings

My major concerns with the validity go back to the methods -- once these are clarified, it will help clarify the results.

Lines 129-131: clarify if the % is based on number of hits or number of total bases. I.e., Firmicutes might have 1000 contigs of 500 nt each; while Proteobacteria might have 10,000 contigs of 200 nt each. Was not clear from methods how the percent was done or typical contig sizes.

General comment throughout the text – significant figures! The sequencing accuracy, when you take all of the different biases and factors into account, is not such that you can go to the hundredths place. It’s also not meaningful conceptually. 67% vs 23% in the text is much easier to process.

Line 134: was observed previously (consider revising whole sentence to be active instead of passive)
Lines 140-144: what is your hypothesis for the animal origin of Arcobacter in sewage? I’m not sure that’s the most relevant point. Bacteroides and Firmicutes are also very common in animals as well as humans. Consider revising this to include discussion about the general community composition of other sewage systems (suggested refs: Vandewalle, Goetz et al 2012, Liu et al 2015, Ye and Zhang 2013) and the prevalence of Arcobacter in sewage (most of these refs will show this is the case; also any of the Collado/Levican/Figueras papers on Arco’s isolated from sewage)

Lines 145-148: this is good info! Would be nice to have a brief description of how the Albertsen method works in the methods. This also helps clarify my earlier question. Consider moving this info to methods.

Lines 148-150 – again, I would focus on the human aspects rather than animal.

Line 152: comparison to A. butzleri does not occur until later; save mentioning until then. Otherwise it is expected that Fig2 is somehow comparing the two, when that does not seem to be the case.

Fig 2: Can you clarify the significance of why there are three rings for presence of different types of functional genes? It seems that all types are shown in each of the levels. It would be very clear if only AR/VR were shown in one ring, prophage/transposase in another. Also showing the DNA coverage would be good.

Fig 3: levels of expression would be nice to see for the different ARG (maybe in supplement). Although many ARG are constitutively expressed, many are also inducible. It seems odd that every single ARG would be expressed. Looking at the level of expression might help sort this out. This result also makes me think there may be DNA contamination. How does expression compare to a single copy gene that should be constitutively expressed?

Fig 3: A comparison of the number of reads that mapped to the non-Arcobacter sewage ARG is important to know. If Arcobacter spp. make up 39% of the total sequences, but the other 60% of sequences contain 30X the number of ARG, this suggests a very high diversity but low abundance. This should be clarified and discussed.

Lines 170-176: since results and discussion are combined, you should be directly comparing the results of whole sewage vs. arcobacter, not just listing them separately.
Again, confirming that there’s no DNA contamination is really key here (or showing expression of ARG relative to a constitutively expressed gene that would be expected to have a higher level of expression).

Line 190-191 Add an example reference of a genus with low HGT and the expected # of shared genes two species with similar relatedness (16S rRNA identity) would typically have just to help frame this.

Lines 203-205: please clarify – are these “genes” being compared only by gene annotation? Or by actual sequence similarity? If only the former, it would be very interesting and informative to see how similar the sequences of these overlapping HGT ARGs are.
It would also be great to show a figure similar to Fig 2 comparing the synteny of the two genomes, particularly with respect to the arrangement of the ARGs, HGT, prophage, and transposon elements. Pleas consider adding such a figure.

Line 228 – add discussion on Arcobacter virulence – Levican et al 2013 AEM is a very nice study looking at the virulence genes known to occur in different species and also in vitro tests with a human cell line to look at infection potential

Lines 241-243: I don’t think that’s really a valid argument. Wastewater treatment has never claimed to remove all bacteria and certainly not to target removal of ones with specific capabilities. A better statement would be to reference one of the numerous publications that shows that WW treatment selects for ABR organisms (e.g., Czekalski et al 2012, Goni-Urriza et al 2000, Mao et al 2015).

Additional comments

Although I have listed a large number of questions and points to correct, I want to be clear that I very much like the paper and want to see it published. Overall, it is fairly simple but elegantly executed -- asks a specific set of questions and provides a clear answer. My comments, though perhaps annoyingly numerous and picky, are intended to improve a manuscript that I feel is very important and that I hope will be widely appreciated.

I am very excited about this work and the implication of the results. This is part of the reason that I want to make sure that everything reported is absolutely correct and done properly, as I will most certainly want to cite this paper in the future.

---

## Round 0.2 · Minor Revisions

Thank you very much for addressing most of the previous concerns, and I thank the reviewers for investing their valuable time once again to evaluate your revised submission.

Both reviewers acknowledge the major improvements in your revised manuscript. Although, as you will see, there are some remaining points that require more attention.

I agree with the points Dr. Speth raise regarding the assessment of the draft genome quality, and his suggestions regarding the gene expression level calculations. Some of those points can be address by including statements that remind the reader what are the potential biases and/or pitfalls of the decisions (i.e. the limited utility of single-copy genes to predict the completion of your genome due to the phylogenetic binning strategy). Others need additional minor work (such as providing clear visualizations to address Dr. Speth's following point).

I am returning your submission with minor revisions, and will be looking forward to your improved submission.

Best wishes,

·

Basic reporting

see below

Experimental design

see below

Validity of the findings

see below

Additional comments

In their revised manuscript, Millar and Raghavan have adressed some of the concerns raised by the reviewers. The methods used have been described more clearly, and in my opinion the manuscript has improved. However, I am not entirely satisfied with some of the changes made to the manuscript. I thus suggest some additional revisions be made before the manuscript is published.


Draft genome quality:
I disagree with the authors that detection of most of the single copy marker genes adresses the problem of preselecting Arcobacter contigs. By definition, the single copy marker genes will be in the Arcobacter core genome, which is specifically selected for using the phylogenetic binning strategy. The (potentially) novel parts of the pan genome are what is discriminated against using your approach, and absence of such regions will not be picked up by any marker gene based completeness assessment.

The authors state they have employed the methods by Albertsen et al. Using these, it should be fairly straightforward to provide a scatterplot of the coverage and GC content of the contigs. If the A cryaerophilus genome is as abundant as it appears from the manuscript that should provide a clear visual way to ascertain that all contigs belonging to the arcobacter genome have been included.

The coverage profile shown in figure 2 shows several large (up to ~10kb) regions with very low coverage to be included in the draft. I assume these are binning errors, and it is unclear to me how they persisted if a round of coverage based binning was performed.

Also with regard to figure 2: In the methods section (line 117) the authors mention "a final scaffold" of 456 contigs, but figure 2 seems to be divided in 14 equal parts with no contig boundaries. What is the meaning of the division of the draft in 14 parts?



Gene expression level:
I have strong doubts with the arbitrary cutoff of 10 reads to claim expression for a gene. The expression level varies more than 1000-fold between the different antibiotic resistance genes, and it is unclear to me why the authors choose to lump all ARGs as expressed.

Considering the 1000fold expression level difference between ARGs, what do the authors mean when they state "housekeeping genes were expressed 10x higher than ARGs"? Were 100 reads mapped to housekeeping genes, or 200,000?

There are a number of published measures for gene expression level that normalize for number of reads mapped, and the length of the gene involved, to actually compare expression levels between genes, such as the reads mapped per kilobase of gene per million reads mapped (RPKM). If the authors want to state expression levels of the ARGs I'd suggest using for example the RPKM and then normalize to one (or multiple) housekeeping genes. Using this to state the levels of ARG expression relative to average expression seems more correct to me. (I suspect that using the current metric, nearly all arcobacter cryaerophilus genes would be considered expressed)

Please do not take these remarks as general criticisms on the work presented. I do think the study is of interest and worth publishing, and PeerJ seems like a suitable outlet to do so. I thus hope the authors will make an effort to address the issues raised here

·

Basic reporting

A few minor grammar errors (mostly subject-verb agreement), but assuming that will be taken care of by copy editors...

Experimental design

No comment.

Validity of the findings

No comment.

Additional comments

The authors appear to have addressed all of my and the other reviewer's comments sufficiently. I am satisfied with the improved manuscript.

---

## Round 0.3 · accepted · Accept

Dear authors,

Thank you very much for your efforts to address Dr. Speth's final points.

Best wishes,